# Rapid LAMP-Based Detection of Mixed Begomovirus Infections in Field-Grown Tomato Plants

**DOI:** 10.3390/v18010019

**Published:** 2025-12-23

**Authors:** Yoslaine Ruiz-Otaño, Berenice Calderón-Pérez, Rosabel Pérez Castillo, Beatriz Xoconostle-Cázares, Alejandro Fuentes Martínez

**Affiliations:** 1Departamento de Biotecnología de las Plantas, Centro de Ingeniería Genética y Biotecnología, Av. 31e/158 y 190, Playa, Apdo, 6162, Habana 10600, Cuba; yoslaine.ruiz@cigb.edu.cu (Y.R.-O.); rosabel.perez@cigb.edu.cu (R.P.C.); 2Departamento de Biotecnología y Bioingeniería, Centro de Investigación y de Estudios Avanzados, Av. Instituto Politécnico Nacional 2508, San Pedro Zacatenco, Ciudad de México 07360, Mexico; bcalderon@cinvestav.mx (B.C.-P.); bxoconos@cinvestav.mx (B.X.-C.); 3Doctorado en Ciencias en Desarrollo Científico y Tecnológico para la Sociedad, Centro de Investigación y de Estudios Avanzados, Av. Instituto Politécnico Nacional 2508, San Pedro Zacatenco, Ciudad de México 07360, Mexico

**Keywords:** LAMP, mixed infections, begomoviruses, tomato mottle Taino virus, tomato latent virus, tomato yellow leaf curl virus, agro-infection

## Abstract

Phytopathogenic viruses severely impact major crops, leading to significant social and economic losses. Among them, begomoviruses pose a serious threat to key cultivars in subtropical and tropical regions despite ongoing efforts to limit their spread. Early detection of these pathogens in field crops and associated weeds is essential for the timely implementation of management strategies to mitigate viral disease outbreaks. In this study, we applied a sensitive loop-mediated isothermal amplification (LAMP) assay for the detection of tomato yellow leaf curl virus (TYLCV), tomato latent virus (TLV), and tomato mottle Taino virus (ToMoTV) in agro-inoculated *Nicotiana benthamiana* and *Solanum lycopersicum*. Importantly, LAMP assays also enabled the identification of these viruses in both symptomatic and asymptomatic field-grown tomato plants, detecting a higher number of infected plants than dot blot hybridization and PCR. Field surveys further revealed mixed infections of TYLCV, TLV, and ToMoTV within individual tomato plants, uncovering a complex epidemiological scenario.

## 1. Introduction

Phytopathogenic viruses severely impact global crop productivity, leading to substantial social and economic losses. Among them, members of the genus *Begomovirus* (family *Geminiviridae*) pose a major threat to agriculture in tropical and subtropical regions despite ongoing efforts to limit their dissemination. Early and accurate detection of these pathogens in cultivated plants and associated weeds is essential for implementing timely management strategies and reducing viral spread. The *Begomovirus* genus comprises single-stranded circular DNA viruses encapsidated in characteristic twinned icosahedral particles. Based on genome organization, begomoviruses are classified as monopartite or bipartite. Monopartite begomoviruses possess a single genomic component (DNA-A), whereas bipartite viruses contain two components (DNA-A and DNA-B). The DNA-A component encodes proteins responsible for replication (AC1/Rep and AC3/REn), transcriptional regulation and silencing suppression (AC2/TrAP), pathogenicity (AC4), and encapsidation (AV1/CP). The DNA-B component encodes proteins essential for intra- and intercellular movement (BV1/NSP and BC1/MP). Both components share a conserved common region that includes the origin of replication. These viruses are transmitted by whiteflies of the *Bemisia tabaci* species complex, which contributes to their rapid global dissemination. Tomato yellow leaf curl virus (TYLCV) is one of the most economically important monopartite begomoviruses, first molecularly identified in Cuba in the late 1990s [1,2]. In Cuba, TYLCV has been prevalent in tomato fields for a long time since the late 1980s, and traditionally cultivated varieties such as Campbell 28 were discontinued due to their high susceptibility to this virus [1,2,3,4]. However, other begomoviruses have been found alongside in tomato fields either alone or mixed with TYLCV, tomato mosaic Havana virus (ToMHV; Y14874) [5], tomato mottle Taino virus (ToMoTV; AF012300) [6] and tomato yellow leaf distortion virus (ToYLDV; FJ174698) [7]. Additionally, TYLCV was detected in combination with tomato chlorotic virus (ToCV) crinivirus [8] and tomato chlorotic spot virus tospovirus (TCSV) [9]. These data challenge the previous belief that TYLCV was predominant. In this complex scenario, a new begomovirus species, tomato latent virus (TLV) (KM926624) [10], was recently identified.

The first demonstration of TLV propagation was obtained through a high-resolution Northern blot hybridization assay and deep sequencing of small viral RNAs in samples of TYLCV-immune plants grown in the field [10]. Furthermore, TLV was tracked by sequencing the RCA product of these tomato plants. The TLV genome sequence exposes a 1080-nucleotide region, spanning the ori, V2, and V1 genes, which is 98.64% similar to the analogous region in the TYLCV genome (KM926626). Furthermore, the TLV genome exhibits 86% similarity to two other weed-infecting species, CoYSV (DQ875868.1) identified in Yucatán, Mexico, and SiGMoV (GU997691.1) identified in FL, USA. These similarities support the likely origin of TLV, which could be associated with a recombination event between TYLCV and a species closely related to CoYSV or SiGMoV [10]. Moreover, of the begomoviruses described in Cuba, ToMoTV showed 75.7% identity with the left region of the TLV ori segment, the highest value. A more detailed intergenic region (IR) analysis revealed the presence of three Rep protein binding sites (iterons), two of which have very similar sequences (AATTGGGGG and AACTGGGGG) [11] located side by side and adjacent to the TATA box, near the C1 gene side. The third, with the inverted sequence CCCCCCTTA [11], is located near the transcription start site of the C1 gene. These iteron sequences were also found in ToMoTV, exhibiting just few base substitutions. Furthermore, the iteron recognition domain (IRD) of the TLV Rep is identical to that of ToMoTV, which is consistent with the similarities found between the iterated sequences in these species.

Intriguingly, TLV was not found to cause symptoms in field-grown tomato plants, unlike ToMoTV, which has a severe impact on the tomato phenotype. In fact, tomato and benthamiana plants agro-infected with TLV in a controlled environment also remain asymptomatic, while ToMoTV induced a symptomatic phenotype just 10 days after agro-infection of tomato seedlings (these results have been included in a second manuscript). These observations further motivate new studies on their spread and pathogenicity in the island as well as in surrounding regions.

Diagnostic methods for begomoviruses have evolved from symptom-based identification to molecular techniques capable of detecting viruses in asymptomatic reservoirs. Conventional approaches such as PCR and DNA hybridization offer high specificity but are time-consuming, require laboratory infrastructure, and depend on skilled personnel [12]. More recently, loop-mediated isothermal amplification (LAMP) has emerged as a promising alternative. This technique enables DNA amplification under isothermal conditions using multiple primers that enhance specificity and efficiency. LAMP is rapid, cost-effective, and suitable for on-site detection, making it a powerful tool for routine diagnostics and surveillance.

LAMP assays have been successfully applied to detect various begomoviruses, including TYLCV, under both laboratory and field conditions [13,14,15,16,17]. Furthermore, commercial LAMP kits targeting conserved regions of the TYLCV genome are available [18], allowing colorimetric visualization or real-time fluorescence-based detection. In the case of ToMoTV and TLV, the development of widely used diagnostic and viral tracking technologies has lagged behind that of TYLCV. Here, we describe the development and evaluation of specific LAMP assays for the detection of TYLCV, ToMoTV, and TLV. Using newly designed primer sets targeting non-conserved viral regions, we assessed assay performance in *N. benthamiana* and tomato plants agroinoculated with infectious clones, and validated their application for detecting single and mixed infections in field-grown tomato crops. This study provides a sensitive, specific, and field-deployable molecular tool to support epidemiological surveillance and integrated management of tomato-infecting begomoviruses.

## 2. Materials and Methods

### 2.1. TYLCV, TLV and ToMoTV Infectious Clones

The dimeric genome of tomato yellow leaf curl virus (TYLCV; accession number AJ223505) cloned into the binary vector pGJ357 was described previously [19]. Tomato latent virus (TLV; KM926624) infectious clone was constructed as a partial dimer in pCambia3300 [10]. Partial dimers of tomato mottle Taino virus (ToMoTV) DNA-A and DNA-B (accession numbers AF012300 and AF012301, respectively) were cloned into pGJ357 and reported earlier [20]. All binary plasmids were introduced into *Agrobacterium tumefaciens* strain LBA4404 by the freeze–thaw method [21] for use in agroinoculation assays [22].

### 2.2. Plant Material and Growth Conditions

*N. benthamiana* plants were cultivated under controlled growth conditions (24 °C, 16 h light/8 h dark photoperiod) with authorization LH47-L (119) from the National Center for Biological Safety (Cuba). Plants were used at the four-leaf stage (approximately 30 days). Tomato (*S. lycopersicum* cv. Campbell-28) seeds were germinated under the same conditions and used at the first true-leaf stage for inoculation experiments. For DNA extraction, apical leaves were harvested and processed using the CTAB method [23].

### 2.3. Virus Inoculation

Agroinoculation of *N. benthamiana* was performed by syringe infiltration of the adaxial leaf surface, as described previously [10]. Tomato plants were inoculated by agroinoculation of wounded epicotyls following the method of [24]. Typical TYLCV and ToMoTV symptoms (yellowing, stunting, and vascular distortion) were observed 10–14 days post-inoculation, whereas TLV-inoculated plants remained asymptomatic after 14 days.

### 2.4. LAMP Assays

Primers were designed with Primer Explorer v.4 (http://primerexplorer.jp, accessed on 22 December 2025). Each primer set (Table 1) consisted of outer (F3, B3), inner (FIP, BIP), and loop primers (F2, F1c, B2, B1c). LAMP reactions were performed in 12.5 µL volumes containing: 0.2 µM each of F3 and B3, 1.6 µM each of FIP, BIP, F2, and F1c, 0.8 µM each of B2 and B1c, 1.4 mM dNTPs (Thermo Fisher Scientific, Waltham, MA, USA), 8 mM MgSO_4_, 1X isothermal amplification buffer, 8 U Bst DNA polymerase (New England Biolabs, Ipswich, MA, USA), and 100 ng of total plant DNA. Parallel assays were performed with 1X WarmStart Colorimetric LAMP Master Mix (New England Biolabs) containing phenol red. Reactions were incubated at 65 °C for 60 min and terminated at 80 °C for 5 min. Products were analyzed by electrophoresis on 2% agarose gels stained with ethidium bromide. Colorimetric LAMP was interpreted as negative if the initial red color remained without change, and positive if the reaction color became yellow, due to the acidification produced by DNA synthesis [25]. Specificity of each primer set was tested against DNA from *N. benthamiana* plants infected with different begomoviruses (TYLCV, ToMoTV or TLV, as needed).

### 2.5. Dot Blot Hybridization

Dot blot hybridization was performed as described previously [25]. PCR amplicons generated with F3–B3 primers were labeled with [α-^32^P] dATP using a random priming kit (Promega, Fitchburg, WI, USA) and used as probes on blotted plant DNA (100 ng). Hybridized membranes were then exposed to X-ray film at −70 °C for four days, revealed and scanned for further interpretation.

### 2.6. PCR Assays

PCR was carried out using 100 ng of total plant DNA in 12.5 µL with virus-specific F3 and B3 primers for TYLCV, TLV, and ToMoTV. Cycling conditions were: 95 °C for 3 min; 35 cycles of 95 °C for 1 min, 65 °C for 1 min, and 72 °C for 30 s; and a final extension at 72 °C for 5 min. Each PCR product was cloned on pDRIVE TA-vector (Qiagen, Santa Clarita, CA, USA) and sequenced. Recombinant plasmids were purified and concentration was adjusted to 1 ng/mL, to be then employed as positive controls when necessary.

### 2.7. Field Trials

For field assays, Campbell-28 seeds were germinated in a 1:1 mixture of peat moss (Terraplant 2; COMPO GmbH & Co. KG, Barcelona, Spain) and zeolite (Litosand; Empresa Geominera del Centro, Santa Clara, Cuba). Plants were maintained in pots under greenhouse conditions until the four-leaf stage and then transplanted to field plots (coordinates: 23°04′29.4″ N, 82°27′11.3″ W). A total of 25 plants were transplanted (15 and 10 per row) in September 2018 and cultivated under natural light and temperature conditions with standard irrigation practices. No pesticides or agrochemicals were applied during the three-month trial (September–November 2018).

## 3. Results

### 3.1. Evaluation of LAMP Primers Using Template DNA from Agroinoculated N. benthamiana and Tomato Plants

LAMP-specific primer sets were designed to target non-conserved viral regions (Table 1). Primers spanning conserved regions of the begomovirus family were excluded to avoid cross-amplification. The selected primers targeted the C3 ORF of TYLCV, the AV1 ORF of the ToMoTV DNA-A component, the BC1 ORF of the ToMoTV DNA-B component, and the C1 ORF of TLV (Table 1). To evaluate the targeted genome regions, PCR was performed using DNA samples from agroinoculated *N. benthamiana* plants with the F3–B3 primer pairs. Amplicons of ~200 bp were obtained as the major products for most PCR detections (Figure 1, lanes 1, 5, 7). For the ToMoTV DNA-A component, three amplicons were observed (Figure 1, lane 3). Sequence analysis confirmed 100% identity of the amplicons with the expected ToMoTV DNA-A genomic region. These ~200 bp amplicons were subsequently used as positive controls for LAMP assays and as labeled probes for dot blot hybridization.

LAMP assays were then performed on total DNA from independent *N. benthamiana* plants agroinoculated with each infectious clone. DNA concatamers, appearing as ladder-like patterns, as observed on agarose gels (Figure 2A). Two sets for detecting ToMoTV were assayed, obtaining amplification in both. No amplification was observed in mock-inoculated plants. Similarly, total DNA from two independent agroinoculated tomato plants produced typical LAMP amplification ladders (Figure 2B), while no products were obtained from mock-inoculated plants or no-template controls (NTC).

### 3.2. Specificity of LAMP Assays

The specificity of LAMP detection was evaluated under potential mixed infection conditions. Cross-amplification tests were conducted using DNA from *N. benthamiana* plants infected with different begomoviruses. TYLCV primers were tested against DNA from ToMoTV- and TLV-infected plants; ToMoTV DNA-A and DNA-B primers were tested against DNA from TYLCV- and TLV-infected plants; and TLV primers were tested against DNA from TYLCV- and ToMoTV-infected plants (Figure 3). No cross-amplification was observed when primers were assayed with plants infected with other viruses.

In addition, each primer set was tested with a complex DNA mixture composed of equal amounts of DNA from *N. benthamiana* plants agroinoculated with TYLCV, TLV, and ToMoTV. Each specific primer set successfully amplified its corresponding target (Figure 4). These results confirmed the specificity of the LAMP assays, even in the presence of complex mixtures representative of mixed viral infections.

### 3.3. Detection of TYLCV, ToMoTV, and TLV in Field-Grown Tomato Plants by LAMP

A total of 25 tomato plants (cv. Campbell-28) were grown under field conditions and exposed to natural inoculation by endemic viruliferous whiteflies, without protection measures. Symptomatic phenotypes were recorded in six plants (plants 12, 13, 15, 17, 18, and 19) at 35 days post-transplantation, while the remaining plants remained asymptomatic. DNA from apical leaves was analyzed by dot blot hybridization, PCR, and LAMP assays (Figure 5). For TYLCV, dot blot assays identified 10 positive plants (plants 4, 9, 11, 12, 15–20). PCR using TYLCV-specific F3–B3 primers confirmed 8 of these detections, but failed to amplify DNA from plants 4 and 20. LAMP, however, detected TYLCV in all dot blot–positive plants as well as in 11 additional plants (1, 2, 5–8, 10, 13, 14, 21, 22, and 24). No detection was obtained in plants 3, 23, and 25 by any method (Figure 5). For ToMoTV, dot blot assays detected four positive plants (4, 11, 12, and 19). PCR detected ToMoTV DNA-A in 13 plants (1–4, 6–7, 9, 13, 15–19, 21–24). LAMP assays confirmed all dot blot- and PCR-positive samples and additionally identified ToMoTV in plants 10, 14, and 25, bringing the total to 21 LAMP-positive plants. Notably, plants 11 and 12, confirmed by dot blot and LAMP, were not detected by PCR. For TLV, dot blot hybridization detected a single positive plant (19). PCR identified TLV in four plants (2, 7, 17, and 19). In contrast, LAMP detected TLV in eight plants (2, 7, 11–14, 17, 19, and 24). Overall, LAMP assays consistently detected more positive samples than PCR or dot blot, including in asymptomatic plants. Mixed infections were common: ten plants and nine plants infected with two and three begomoviruses, respectively. These findings reveal a complex epidemiological scenario in field-grown tomato.

### 3.4. Colorimetric LAMP Detection of TYLCV, ToMoTV and TLV in Agro-Infected Tomato Plants

Based on the built capacity to specifically detect the viruses, colorimetric LAMP was assayed, considering it brings simplicity and portability for screening on-site. Figure 6 shows the colorimetric LAMP detection of TYLCV, ToMoTV and TLV in agro-infected tomato plants, employing systemic leaves. The upper panel shows the yellow color associated with the DNA in two representative infected plants. The amplified DNA products were then resolved in an agarose gel, thus confirming the presence of the expected ladder produced by LAMP concatamers. NTC-labeled tubes conserved the red, initial color, indicating no acidification of the reaction for lack of synthesis of each viral gene target.

### 3.5. Limit of Detection (LOD) of the Colorimetric LAMP of TYLCV, ToMoTV and TLV

Recombinant plasmids containing defined copy numbers of target genes from TYLCV, ToMoTV and TLV were spiked with total DNA from healthy plants and colorimetric LAMP was performed. LOD was defined as observed true positives when a yellow color shift was observed. DNA dilutions from 10^2^ to 10^6^ copies of target genes were assessed for each virus. Non-Template Control (NTC) remained red, while LOD of 10^5^ copies per reaction for TYLCV and ToMoTV was observed, whereas TLV was more sensitive, detecting 10^4^ copies per reaction (Figure 7). Although faint LAMP products can be observed in 10^3^ and 10^2^ copies tested, it was not enough to shift the color; yellowish color was interpreted as negative.

## 4. Discussion

Early and accurate detection of plant viruses is essential for implementing timely and specific control strategies [12]. Otherwise, new viral outbreaks in crop plants may only be recognized once symptoms have become evident and disease has progressed. Viruses often occur in single or mixed infections within individual plants, and when co-infecting, they can interact either synergistically or antagonistically. In synergistic interactions, symptoms tend to be more severe, frequently leading to substantial yield losses [25,26,27,28,29]. In addition, viruses may circulate among asymptomatic plants, interfering with or complementing each other’s infection cycles, thereby serving as latent sources of future outbreaks [29,30]. Therefore, the availability of a rapid and reliable diagnostic tool for early virus identification is crucial to prevent viral spread and to minimize virus–virus and virus–host interactions.

In this study, we demonstrated the suitability of LAMP assays for the detection of tomato yellow leaf curl virus (TYLCV), tomato mottle Taino virus (ToMoTV), and tomato latent virus (TLV) in both *N. benthamiana* and tomato plants. The performance of the LAMP assay was first evaluated using the model host *N. benthamiana*, agroinoculated with infectious clones of each virus. This species is a well-established experimental system for plant virology because it supports replication of a wide range of viruses and can be easily maintained under controlled conditions [31]. Consistent with this, high replication levels were achieved for TYLCV, ToMoTV, and TLV in *N. benthamiana*. In contrast, agroinoculated tomato plants displayed variable viral titers, despite being the natural host species from which these begomoviruses were originally isolated. Such differences may reflect host-dependent responses influencing viral replication, movement, or suppression. Nonetheless, all agroinoculated samples were successfully detected as positive by LAMP, demonstrating the robustness of the assay even at potentially low viral concentrations. Indeed, the colorimetric detection allowed a rapid and specific detection just by observing the color of the reaction, turned to yellow in those positive samples. Furthermore, the high specificity of the LAMP primers enabled accurate detection of each virus in mixed DNA preparations, confirming their discriminatory capacity among TYLCV, TLV, and ToMoTV. Importantly, for TYLCV we designed new LAMP primers targeting the C2–C3 overlapping region, distinct from previous primer sets [18] that targeted the intergenic region (IR), C4, and V2/V1 ORFs. Our new primer set provided specific amplification without cross-reactivity to ToMoTV or TLV, and generated consistent ladder-like amplicons. This adds a valuable alternative design for TYLCV detection and expands the available primer repertoire for LAMP-based begomovirus diagnostics. For the case of TYLCV, LAMP proved useful for laboratory testing as well as for field diagnosis, even without the need for DNA extraction, using the immunocapture procedure followed by the DNA amplification reaction [31], or using enhanced detection of viral amplicons by a Cas12-directed fluorescent reporter [32]. A diagnostic kit is also available from NIPPON GENE CO., LTD (Toyama, Japan), which includes primers targeting a conserved region of the TYLCV genome, enabling the detection of a broad range of TYLCV strains, based on a previous key publication [33]. Its application also allows for rapid visualization of the genome’s DNA by colorimetric methods [31], or by combining the LAMP assay with real-time methods, its sensitivity could be improved [30]. In all these examples, the potential and flexibility of the LAMP procedure have been demonstrated.

Similarly, new specific primers were developed for ToMoTV and TLV, allowing their simultaneous or independent detection with high sensitivity. Colorimetric LAMP detection couples thermal amplification with a simple color change that signals the presence of viral nucleic acid with no instruments required. That makes it ideal for field and resource-limited settings, as a result is obtained in 40 min, only with the use of a heat block with constant 60 °C, being highly specific and sensitive enough to detect infected plants. For future use, room-temperature lyophilized mixes for on-site use will ease cold-chain constraints, and the color change is intuitive for extension workers and growers.

Recombinant plasmids containing defined copy numbers of target genes from TYLCV, ToMoTV and TLV were spiked with total DNA from healthy plants and colorimetric LAMP was performed. LOD was defined as observed true positives when a yellow color shift was observed. DNA dilutions from 10^2^ to 10^6^ copies of target genes were assessed for each virus. Non-Template Control (NTC) remained red, while LOD of 10^5^ copies per reaction for TYLCV and ToMoTV was observed, whereas TLV was more sensitive, detecting 10^4^ copies per reaction (Figure 7). Although faint LAMP products can be observed in the agarose gel for 10^3^ and 10^2^ copies tested, it was not enough to shift the color; tubes with yellowish color were interpreted as negative.

Although the LAMP diagnostic tool is sensitive and enables early detection of target nucleic acids, the mandatory use of both positive and negative controls is strongly recommended. Field use of portable devices may lead to false-positive results, likely due to environmental contamination. Therefore, any positive result obtained from a field sample should be confirmed in the laboratory using good laboratory practices (e.g., decontaminated work surfaces and dedicated, decontaminated pipettes).

Field-grown tomato plants were confirmed to be infected with TYLCV, ToMoTV, and TLV, even though most plants were asymptomatic, likely reflecting early infection stages or complex virus–host–environment interactions modulating symptom expression. The co-occurrence of TYLCV, ToMoTV, and TLV within the same field, and in some cases within the same individual plants, is particularly significant, marking the first report of this association.

Mixed begomovirus infections have been widely documented in various regions and hosts. Several bipartite and monopartite begomoviruses have been found co-infecting common weeds. For instance, *Desmodium leaf distortion virus* (bipartite) and *Corchorus yellow vein Cuba virus* (monopartite) were detected together in *Corchorus siliquosus* [34]. Such associations can become even more complex through interactions with betasatellites or deltasatellites, which may enable pseudo-recombination between mono- and bipartite genomes [35], thereby facilitating viral diversification, spread, and symptom modulation. Globally, mixed infections between Old World monopartite and New World bipartite begomoviruses have been reported in crop species [10,35,36,37,38,39,40], and the transfer of bipartite viruses from weeds to cultivated plants has also been demonstrated [36]. However, in the region, natural co-infections of monopartite and bipartite begomoviruses in tomato crops had not been previously confirmed.

Our findings reveal the natural association of TYLCV (monopartite), ToMoTV (bipartite), and TLV (monopartite) in tomato, which is epidemiologically relevant. Shared genomic features may underlie their ability to co-infect the same host. TYLCV and TLV share similar V1 and V2 ORFs, with the latter known to act as a suppressor of RNA silencing [41], potentially modulating antiviral responses and facilitating co-infection. Additionally, similarity in iteron motifs between ToMoTV and TLV may allow replication cross-compatibility or pseudo-recombination. Such interactions could enhance viral adaptability or symptom variability in co-infected plants. Nevertheless, further studies are needed to confirm the viability of these associations and to identify whether nearby weeds serve as viral reservoirs facilitating their transmission. Historically, TYLCV has predominated in tomato fields in the region since its first detection in the late 1980s, as repeatedly confirmed in field surveys. However, our current findings indicate that tomato crops now harbor complex mixed begomovirus populations, with no clear dominance of the typical TYLCV-associated disease phenotype. Whether this shift results from virus–virus synergism, competition, or altered plant–vector dynamics is still an open question. The hidden mechanism in these mixed infections can be revealed in the near future under controlled viral inoculation conditions using infectious clones. The emergence of these mixed infections may reflect evolving agroecosystem conditions and intensified agricultural practices, which could favor viral dissemination among neighboring crops and associated weed communities [42]. Given the reality of the more frequent occurrence of mixed infections alongside environmental changes and their effects on living organisms, the feasibility of developing and implementing multiplex LAMP is being promoted [43].

## 5. Conclusions

The present study demonstrates that the loop-mediated isothermal amplification (LAMP) assay provides a rapid, sensitive, and highly specific diagnostic tool for the detection of tomato yellow leaf curl virus (TYLCV), tomato mottle Taino virus (ToMoTV), and tomato latent virus (TLV) in both experimental and field conditions. The newly designed primer sets enabled accurate identification of single and mixed begomovirus infections, even in asymptomatic plants, proposing a tool to overcome the use of conventional PCR and dot blot hybridization diagnosis. Its use as colorimetric LAMP assays is a suitable tool for on-site detection, as no sophisticated equipment is required which marks it as an additional advantage. The simultaneous detection of TYLCV, ToMoTV, and TLV in field-grown tomato plants reveals a complex and evolving epidemiological landscape in tomato agroecosystems and indicates that new approaches to reduce false negatives that may appear due to overexposure of viral targets in one sample would be important for accurate diagnosis. In the current scenarios of mixed infections, the implementation of the use of multiplex LAMP in the nearest future would increase the applicability of the presented proposal. The described findings highlight the relevance of LAMP for routine surveillance, providing a valuable platform for early virus monitoring and for guiding integrated management strategies to mitigate the spread and impact of begomovirus-associated diseases.

## Figures and Tables

**Figure 1 viruses-18-00019-f001:**
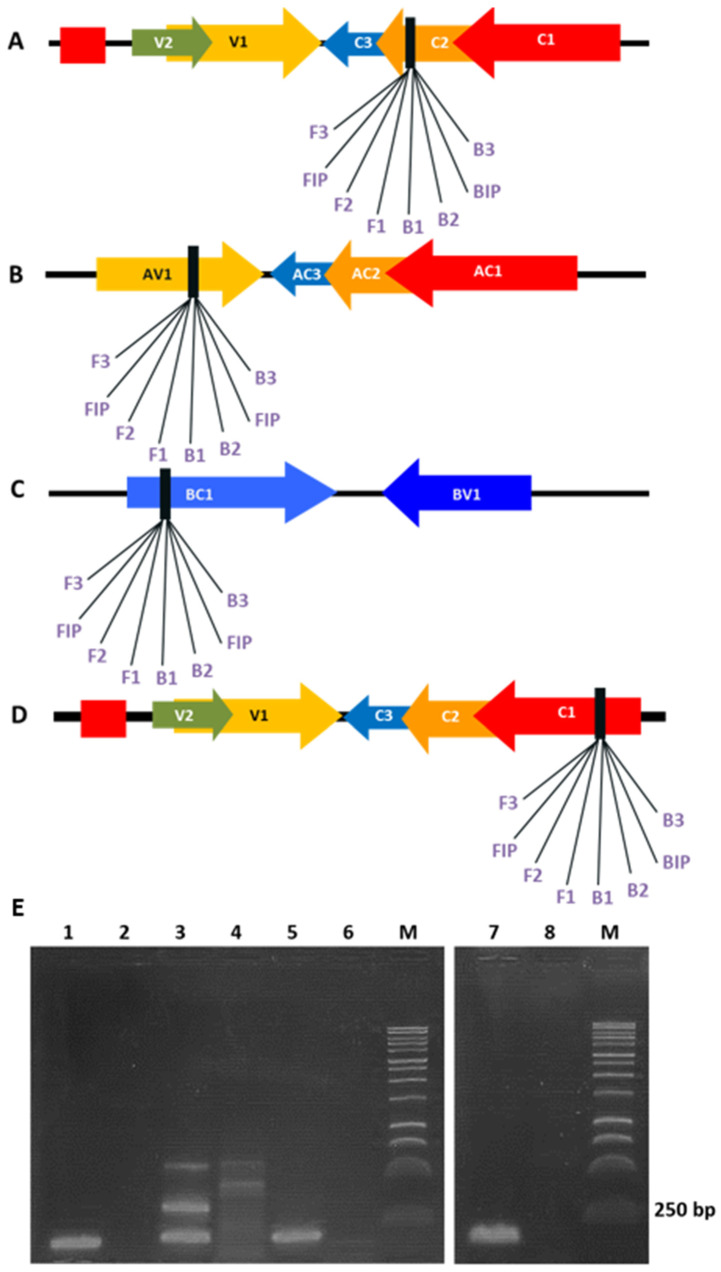
Depicted LAMP primer sets for TYLCV, ToMoTV and TLV detection. Left panel: open reading frames (ORFs) from viral genomes are represented by arrows, and designed primers for the LAMP assays are placed on target regions. (**A**) Primers spanning C2-C3 ORF of TYLCV genome; (**B**) Primers spanning AV1 ORF of ToMoTV DNA-A component; (**C**) Primers spanning BC1 OFR of ToMoTV DNA-B component; (**D**) Primers spanning C1 ORF of TLV genome. (**E**) Agarose gel electrophoresis of PCR products using total DNA samples from agro-infected *Nicotiana benthamiana* plants and F3-B3 primers of each specific set. Lanes 1: TYLCV detection; 3: ToMoTV DNA-A detection; 5: ToMoTV DNA-B detection; 7: TLV detection; 2, 4, 6, 8: mock-inoculated plants; M: 1 kb DNA ladder (Promega).

**Figure 2 viruses-18-00019-f002:**
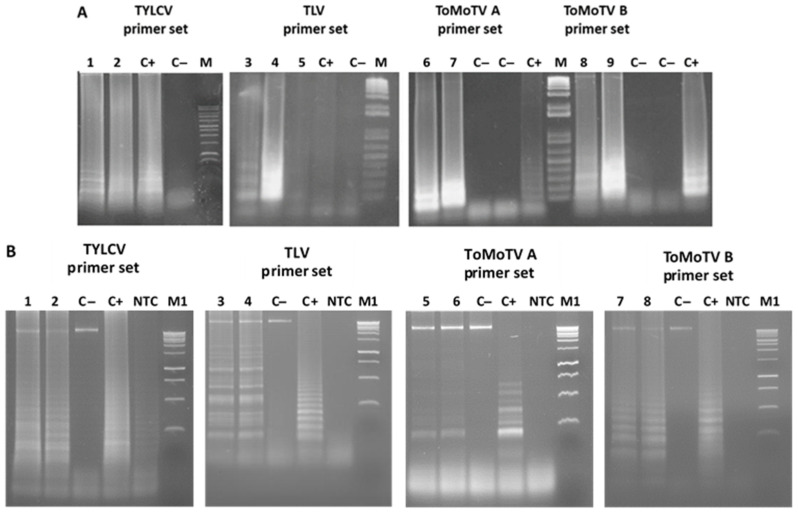
LAMP amplification of total DNA from agro-infected plants for virus detection. Agarose gel electrophoresis of LAMP products for TYLCV, TLV and ToMoTV detection. Total DNA from agro-infected plants was used with specific primer set for each virus as indicated. (**A**) *Nicotiana benthamiana* plants; (**B**) tomato plants. Lanes 1–2: TYLCV-infected plants; 3–5: TLV-infected plants; 6–9: ToMoTV-infected plants. C−: Mock-inoculated plants; C+: positive controls; NTC: non-template control. M: 1 kb DNA ladder (Promega); M1: 1 kb Plus DNA Ladder (Invitrogen, Waltham, MA, USA).

**Figure 3 viruses-18-00019-f003:**
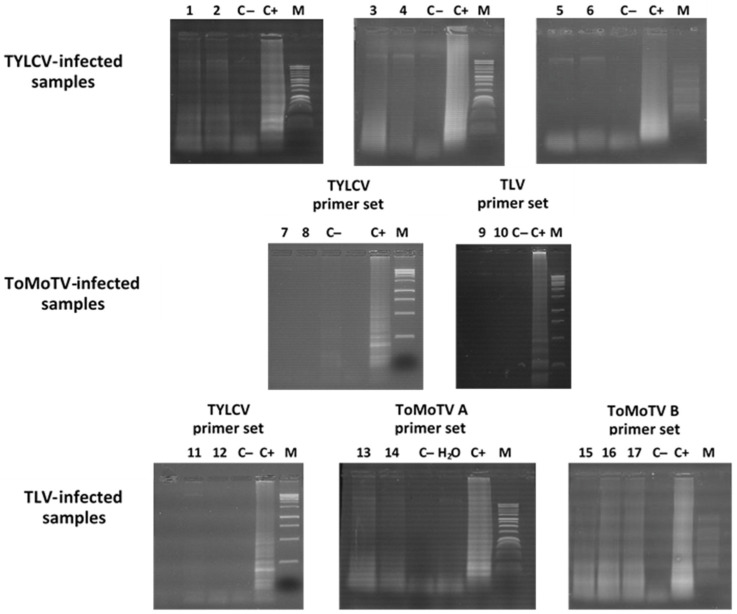
Detection specificity of LAMP assays. Agarose gel electrophoresis of LAMP products of crossed detections. Each primer set was assessed in LAMP reactions with total DNA from agro-infected *Nicotiana benthamiana* plants. Lanes 1–6: TYLCV-infected samples; 7–10: ToMoTV-infected samples; 11–17: TLV-infected samples. C−: Mock-inoculated plants. C+: Positive controls corresponding to each set of specific primers. M: 1 kb DNA Ladder (Invitrogen).

**Figure 4 viruses-18-00019-f004:**
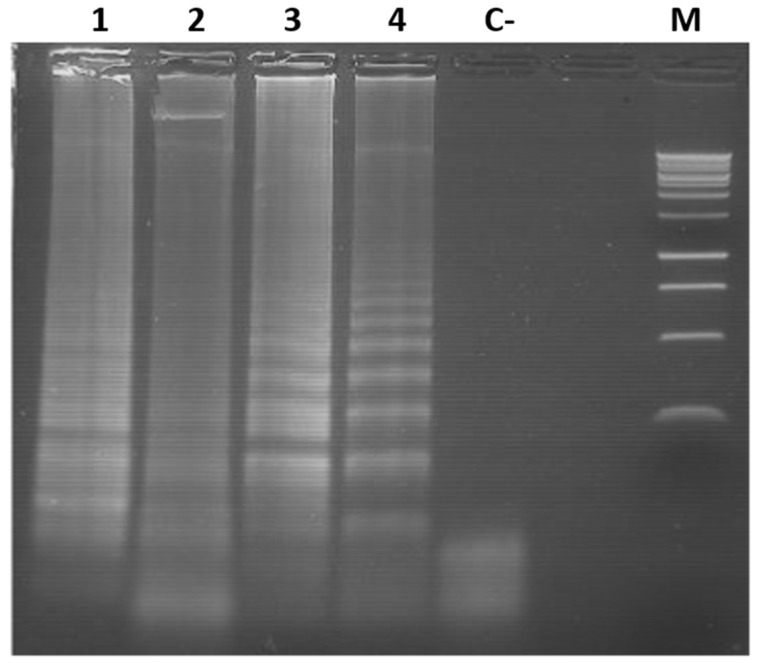
Specific LAMP amplification for virus detection in a complex DNA sample. Agarose gel electrophoresis of LAMP products for TYLCV, ToMoTV and TLV detection in a complex DNA sample consisted in a mixture of equal amounts of total DNA extracted from agro-infected *N. benthamiana* plants. Different primer sets were used for viral detection. Lanes 1: TYLCV primer set; 2: ToMoTV DNA-A primer set; 3: ToMoV DNA-B primer set; 4: TLV primer set. C−: Mock-inoculated plant DNA tested with all primer sets. M: 1 kb DNA Ladder (Promega).

**Figure 5 viruses-18-00019-f005:**
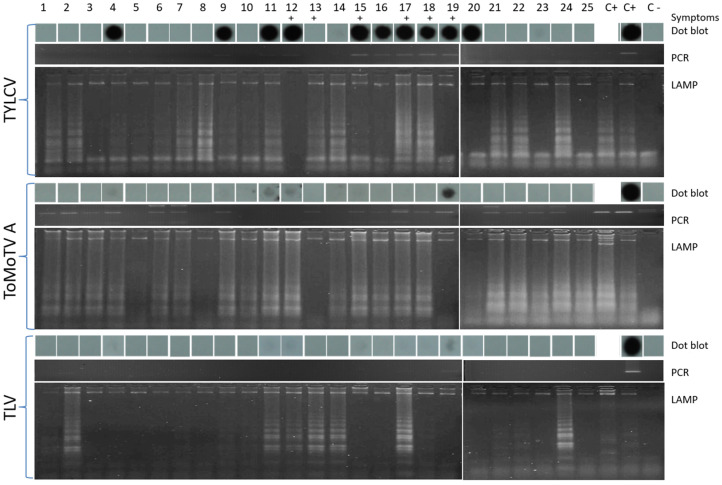
Detection of TYLCV, ToMoTV and TLV in field-grown tomato plants. Dot blot hybridization, PCR and LAMP assays with plant total DNA and specific probes/primers were performed for each begomovirus (TYLCV, ToMoTV DNA-A component and TLV). The symptomatic plants are represented by the symbol (+). The numbers (1–25) correspond to independent field-grown tomato plants as previously described. C−: Mock-inoculated plant. C+: Positive control.

**Figure 6 viruses-18-00019-f006:**
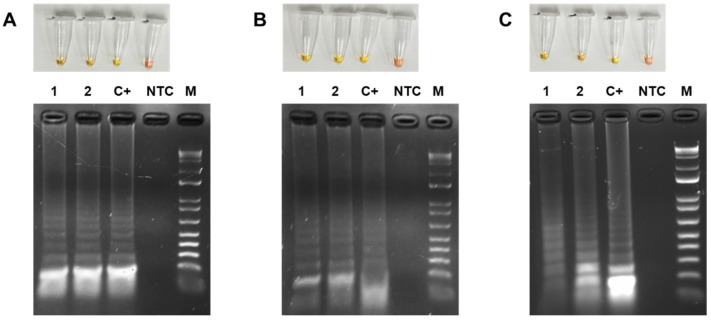
Colorimetric LAMP detection of TYLCV, ToMoTV and TLV in agro-infected tomato plants. Colorimetric LAMP assays (upper panel) and electrophoretic profile of the amplification reaction products (lower panel) for TYLCV (**A**), ToMoTV (**B**) and TLV (**C**) detection. The numbers (1–2) correspond to independent tomato plants infected with each begomovirus. C+: Positive control. NTC: non-template control. M: 1 kb Plus DNA Ladder (Invitrogen). Yellow color, positive; red color, negative.

**Figure 7 viruses-18-00019-f007:**
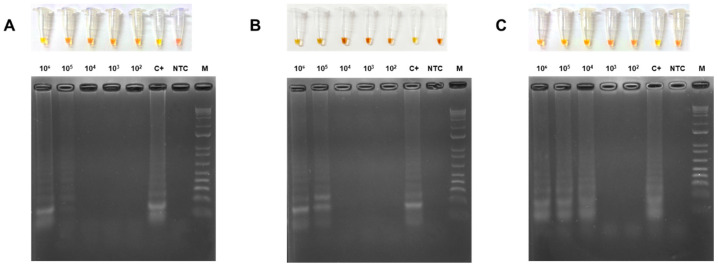
Determination of LOD for colorimetric LAMP assays of TYLCV, ToMoTV and TLV. Colorimetric LAMP assays (upper panel) and electrophoretic profile of the amplification reaction products (lower panel) for TYLCV (**A**), ToMoTV (**B**) and TLV (**C**) detection. Ten-fold serial dilutions of PCR products ranging from 10^6^ to 10^2^ copies/reaction were used. C+: Positive control. NTC: non-template control. M: 1 kb Plus DNA Ladder (Invitrogen). Yellow color, positive; red color, negative.

**Table 1 viruses-18-00019-t001:** Primers designed for LAMP assays.

Virus-Primer	Gene/Position	Sequence 5′-3′
TYLCV-F3	C3/1234-1254	TCTTAAGAAACGACCAGTCT
TYLCV-B3	C3/1409-1432	TTTATCTGGGAGATAATCAATCC
TYLCV-FIP	C3/1280-1328	CCACAACATCAGGAAGGTAATGGGGATTGGCTGTAATGTCGTCCAAAT
TYLCV-BIP	C3/1337-1390	ATCTGAATGGAAATGATGTCGTGGTTCATTCTCTATTTCAAGATAACAGACCA
TYLCV-F2	C3/1256-1276	GGCTGTAATGTCGTCCAAAT
TYLCV-F1c	C3/1305-1328	CCACAACATCAGGAAGGTAATGG
TYLCV-B2	C3/1385-1408	CTCTATTTCAAGATAACAGACCA
TYLCV-B1c	C3/1337-1361	ATCTGAATGGAAATGATGTCGTGG
ToMoTV-A-F3	AV1/563-582	CCACACGAACAGCGTCATG
ToMoTV-A-B3	AV1/757-775	GTCGAACCAGAGCCTGCT
ToMoTV-A-FIP	AV1/619-666	ACTGGGCTCGTTGTCGTACATGTTGAACTTGGCTAGTACGAGACCGG
ToMoTV-A-BIP	AV1/666-707	ACTGCCACTGTGAAGAACGACCTCCGGGCATACTGACCACC
ToMoTV-A-F2	AV1/584-603	TTGGCTAGTACGAGACCGG
ToMoTV-A-F1c	AV1/644-666	ACTGGGCTCGTTGTCGTACATG
ToMoTV-A-B2	AV1/728-748	CATACTGACCACCAGTCACC
ToMoTV-A-B1c	AV1/666-688	ACTGCCACTGTGAAGAACGACC
ToMoTV-B-F3	NSP/413-434	TTTTCATATGACTAACCGACG
ToMoTV-B-B3	NSP/600-621	GCTGATAAACGTTGAGATAGC
ToMoTV-B-FIP	NSP/451-505	CTCCTCGACGTTTTCCATCATGTCGTTTATCTCGTTATTCTATGTTTAACCGTA
ToMoTV-B-BIP	NSP/521-566	CACTGATGAGCCCAAGATGACAGCCCATGAATTATGGGCCAGGAC
ToMoTV-B-F2	NSP/441-466	TCTCGTTATTCTATGTTTAACCGTA
ToMoTV-B-F1c	NSP/482-505	CTCCTCGACGTTTTCCATCATGT
ToMoTV-B-B2	NSP/582-600	TGAATTATGGGCCAGGAC
ToMoTV-B-B1c	NSP/521-542	CACTGATGAGCCCAAGATGAC
TLV-F3	C1/1700-1719	AGCACGATTGAAGGGATAC
TLV-B3	C1/1873-1892	CAGTGGACATCTGGACTTC
TLV-FIP	C1/1747-1794	GGAAAGAACTTCTGGGGGCCCAGAAGCTCCTTTAATTTGAACTGGCT
TLV-BIP	C1/1794-1845	AGTGCTTTAGCTTTAGATAGTGCGGTGCGACTCGAGTCTATTCGAACGAAG
TLV-F2	C1/1719-1740	CTCCTTTAATTTGAACTGGCT
TLV-F1c	C1/1774-1794	GGAAAGAACTTCTGGGGGCC
TLV-B2	C1/1848-1869	CTCGAGTCTATTCGAACGAAG
TLV-B1c	C1/1794-1818	AGTGCTTTAGCTTTAGATAGTGCG

## Data Availability

The original contributions presented in this study are included in the article. Further inquiries can be directed to the corresponding author.

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
