# Peer review of "Rapid LAMP-Based Detection of Mixed Begomovirus Infections in Field-Grown Tomato Plants"

_viruses, 2025, doi:10.3390/v18010019_

Round 1
Reviewer 1 Report
Comments and Suggestions for Authors
Dear authors
Your submission seemed to me very interesting, well developed experiments and of high importance in the development of suitable begomovirus detection tools. I consider that your submission is accepted by me in its present form. Congrats, very nice study.
- The possibility that LAMP methodology is more accurate and suitable than others Was adequately tested.
- By the first timeit is demostrated the effective application of LAMP detecting these GVs. This open the possibility to evaluate this cheaper and fast strategy, in detecting other pathogens.
- In the methodology, to mention briefly in discussion the pros and cons of using LAMP in molecular detection of viruses.
- I only suggest to include pros and cons in conclusions, in a more brief descripción in comparison with discussion.
Best
Author Response
Dear Editor and reviewers,
Thank you very much for the high-quality job you did. We consider to have addressed all your valuable comments that substantially improved the manuscript.
Please, find below to your comments and suggestions our answers and additional text we considered to include to provide the requested information.
Sincerely,
Alejandro Fuentes, on behalf of the authors
Comments and Suggestions for Authors
Dear authors
Your submission seemed to me very interesting, well developed experiments and of high importance in the development of suitable begomovirus detection tools. I consider that your submission is accepted by me in its present form. Congrats, very nice study.
The possibility that LAMP methodology is more accurate and suitable than others Was adequately tested.
By the first time it is demonstrated the effective application of LAMP detecting these GVs. This open the possibility to evaluate this cheaper and fast strategy, in detecting other pathogens.
In the methodology, to mention briefly in discussion the pros and cons of using LAMP in molecular detection of viruses.
I only suggest to include pros and cons in conclusions, in a more brief description in comparison with discussion.
We appreciate very much your opinion on the manuscript.
Answer: The text was modified as follows:
Its use as colorimetric LAMP assays is a suitable tool for on-site detection, as no sophisticated equipment is required which marks it as an additional advantage. The simultaneous detection of TYLCV, ToMoTV, and TLV in field-grown tomato plants reveals a complex and evolving epidemiological landscape in tomato agroecosystems and indicates that new approaches to reduce false negatives that may appear due to overexposure of viral targets in one sample would be important for accurate diagnosis. In the current scenarios of mixed infections, the implementation of the use of accurate multiplex LAMP in the nearest future would increase the applicability of the presented proposal.
Reviewer 2 Report
Comments and Suggestions for Authors
Begomoviruses pose a serious threat to important crops in subtropical and tropical regions. The authors present a LAMP assay for detecting TYLCV, TLV, and ToMoTV, offering a promising approach for early detection and disease management. The work is technically solid, but several key issues need to be addressed to strengthen the validity and broader applicability of the reported results.
ABSTRACT
- Lines 21-23: The virus name should not be capitalized. Please double-check this throughout the text.
- Line 23 “Nicotiana Benthamiana” and “Solanum Lycopersicon” should be written in italic
- Lines 25-26: The claim of superior sensitivity requires quantitative support. The authors state that the LAMP assay demonstrates higher sensitivity than dot blot hybridization and PCR, but they do not provide specific comparative data. To substantiate this claim, it is essential to include quantitative measures of the sensitivity difference, such as the direct comparison of detection limits (e.g., the lowest copy number or concentration detectable by each method).
INTRODUCTION
Major comment:
- Background context is underdeveloped. The introduction provides insufficient detail on the biology and epidemiology of TLV and ToMoTV in Cuba. To better contextualize the study, the authors should include information on the discovery, pathogenicity, and epidemiological significance of these viruses in the region.
- Discussion lacks key comparisons. The discussion would be significantly strengthened by acknowledging and citing existing literature on LAMP-based detection of TYLCV, a closely related virus. The authors must use this comparison to precisely articulate the specific advance or novel contribution of their present work.
- Unsubstantiated claim and internal inconsistency. The claim that "diagnostic tools for ToMoTV and TLV remain unavailable" (Lines 71-72) is not sufficiently supported. The prior mention of TLV detection in Cuban fields and the availability of its complete genome in GenBank contradict this statement, as these facts demonstrate that molecular detection is feasible. The authors must either provide evidence for this claim or revise the text for accuracy.
Minor comments:
- Line 35 ‘Geminiviridae’ should be written in italic.
- Lines 54 -55 The virus name should not be capitalized. Please double-check this throughout the text.
MATERIALS AND METHODS
- Primer Design: The manuscript does not explain the rationale for selecting the C3, AV1, BC1, and C1 ORF regions as amplification targets. In particular, the choice of these relatively non-conserved regions requires justification. The authors should elaborate on the specificity of these regions for virus identification and discuss their degree of sequence variability within the Begomovirus
- Section 2.4 (Field Trials) should be relocated to the end of the M&M section and renumbered as Section 2.7. The preceding subsections should be shifted accordingly.
- Field sample validation: To more robustly validate the reliability of the established LAMP assay, it is recommended to increase the number of field-collected samples tested.
- Template concentration consistency: The template input is inconsistently reported: both PCR and LAMP used 100 ng of DNA, but the total reaction volumes differ (e.g., PCR was performed in 25 µL, whereas the LAMP volume was not specified). This discrepancy leads to a two-fold difference in template concentration, which compromises the fairness of sensitivity comparisons between the two methods. It is recommended that the authors clearly state the template concentration in consistent units (e.g., ng µL⁻¹) and ensure equivalent reaction conditions for an accurate comparison.
RESULTS
- Image clarity and presentation: Figures 2–7 are not sufficiently clear and should be replaced with higher-resolution images. In addition, the electrophoresis lanes currently split across Figures 2–4 should be consolidated into a single composite image. The figure legend should clearly annotate what each lane represents.
- Electrophoresis image quality: The resolution of the electrophoresis images is inadequate. In Figures 2A and 2B, the upper fragments of the LAMP ladder exceed 2 kb, indicating possible overloading or improper separation. It is recommended to repeat the electrophoresis using a DNA ladder more appropriate for this fragment size range.
- Specificity testing is incomplete: The specificity evaluation in Figure 3 is limited to cross-testing with single-infection DNA samples and does not include testing with field samples containing dual or triple infections. This oversight leaves open the possibility of false negatives due to template competition. To address this, the authors should include DNA extracted from a sequenced-confirmed triple-infected field sample as a template in the LAMP assay.
- Limit of detection (LOD) interpretation is overly subjective. The LOD determination in Figure 7 is based solely on visual color change, which is subjective and not corroborated by gel electrophoresis. To improve objectivity, the authors should indicate—either in the figure or the legend—at which template concentrations a band was visible by electrophoresis even in the absence of a color change.
DISCUSSION
- Lack of experimental support for pathogenicity in mixed infections: While the authors speculate that "synergism or competition" may underlie the observed symptom differences, this hypothesis lacks experimental validation through inoculation studies using infectious clones. To strengthen their argument, the authors should explicitly frame this as a proposed mechanism and recommend its verification in future research.
- Absence of comparison with commercially available TYLCV LAMP kits: Although the authors acknowledge the existence of commercial LAMP reagents for TYLCV detection, no direct comparison was made with their newly developed primers. This omission undermines claims regarding the superiority or competitiveness of their method. The authors should either include such a comparative analysis or provide a clear justification for its exclusion (e.g., restrictions on reagent availability in Cuba).
- Underdeveloped discussion of LAMP applicability: The study does not adequately address the potential of LAMP for high-throughput applications or simultaneous detection of multiple viruses. To enhance the impact and relevance of the work, the authors should expand the discussion to include the feasibility of developing multiplex LAMP assays or integrating this method into microfluidic platforms for broader diagnostic use.
REFERENCES
The references are not formatted according to the journal's required style. The authors should carefully revise all citations and the reference list to ensure full compliance with the journal's guidelines.
Lines 204-210
Figure 4 lacks essential annotations: the molecular weight marker is not labeled with "bp," and there is a discrepancy between the lane numbering in the text (referring to lanes 1–4) and the actual figure (labeled 1, 2, 3, 4). The authors should ensure consistency between the figure and its description.
Author Response
Reviewer 2:
Open Review
Dear Editor and reviewers,
Thank you very much for the high-quality job you did. We consider to have addressed all your valuable comments that substantially improved the manuscript.
Please, find below to your comments and suggestions our answers and additional text we considered to include to provide the requested information.
Sincerely,
Alejandro Fuentes, on behalf of the authors
Comments and Suggestions for Authors
Begomoviruses pose a serious threat to important crops in subtropical and tropical regions. The authors present a LAMP assay for detecting TYLCV, TLV, and ToMoTV, offering a promising approach for early detection and disease management. The work is technically solid, but several key issues need to be addressed to strengthen the validity and broader applicability of the reported results.
ABSTRACT
Lines 21-23: The virus name should not be capitalized. Please double-check this throughout the text.
Thank you, names are now in agreement to the International Committee on Taxonomy of Viruses (ICTV), accessed on Dec 13, 2025. Virus species names (taxa) are binomial and italicized, ICTV specifies that a virus species name consists of two words in italics: Word 1 = the genus name (capitalized; identical to the genus spelling); Word 2 = the species epithet (letters from the Latin alphabet and numbers). In contrast, common virus name should be not italic.
https://ictv.global/faq/names "How do I write virus, species, and other taxa names?
Line 23 “Nicotiana Benthamiana” and “Solanum Lycopersicon” should be written in italic. Done, thank you.
Lines 25-26: The claim of superior sensitivity requires quantitative support. The authors state that the LAMP assay demonstrates higher sensitivity than dot blot hybridization and PCR, but they do not provide specific comparative data. To substantiate this claim, it is essential to include quantitative measures of the sensitivity difference, such as the direct comparison of detection limits (e.g., the lowest copy number or concentration detectable by each method).
..demonstrating higher sensitivity- was replaced by detecting a higher number of infected plants than..
INTRODUCTION
Major comment:
Background context is underdeveloped. The introduction provides insufficient detail on the biology and epidemiology of TLV and ToMoTV in Cuba. To better contextualize the study, the authors should include information on the discovery, pathogenicity, and epidemiological significance of these viruses in the region.
Answer: Additional text was included:
“In Cuba, TYLCV has been prevalent in tomato fields for a long time since the late 1980s, and traditionally cultivated varieties such as Campbell 28 were discontinued due to their high susceptibility to this virus (Ramos et al. 1996; Zubiaur et al. 1996; 2002; 2004; Quiñones et al. 2002). However, other begomoviruses have been found alongside in tomato fields either alone or mixed with TYLCV, tomato mosaic Havana virus (ToMHV; Y14874) (Zubiaur et al., 1998), tomato mottle Taino virus (ToMoTV; AF012300) (Ramos et al., 1997) and tomato yellow leaf distortion virus (ToYLDV; FJ174698) (Fiallo-Olivé et al., 2012a). Additionally, TYLCV was detected in combination with tomato chlorotic virus (ToCV) crinivirus (Martínez-Zubiaur et al., 2008) and tomato chlorotic spot virus tospovirus (TCSV) (Martínez-Zubiaur et al., 2016). These data challenge the previous belief that TYLCV was predominant. In this complex scenario, a new begomovirus species, tomato latent virus (TLV) (KM926624) (Fuentes, 2016), was recently identified.
The first demonstration of TLV propagation was obtained through a high-resolution Northern blot hybridization assay and deep sequencing of small viral RNAs in samples of TYLCV-immune plants grown in the field (Fuentes, 2016). Furthermore, TLV was tracked by sequencing the RCA product of these tomato plants. The TLV genome sequence exposes a 1080-nucleotide region, spanning the ori, V2, and V1 genes, that is 98.64% similar to the analogous region in the TYLCV genome (KM926626). Furthermore, the TLV genome exhibits 86% similarity to two other weed-infecting species, CoYSV (DQ875868.1) identified in Yucatán, Mexico, and SiGMoV (GU997691.1) identified in Florida, USA. These similarities support the likely origin of TLV, which could be associated with a recombination event between TYLCV and a species closely related to CoYSV or SiGMoV (Fuentes, 2016). Moreover, of the begomoviruses described in Cuba, ToMoTV showed 75.7% identity with the left region of the TLV ori segment, the highest value. A more detailed intergenic region (IR) analysis revealed the presence of three Rep protein binding sites (iterons), two of which have very similar sequences (AATTGGGGG and AACTGGGGG) (Argüello-Astorga et al., 1994), located side by side and adjacent to the TATA box, near the C1 gene side. The third, with the inverted sequence CCCCCCTTA (Argüello-Astorga et al., 1994), is located near the transcription start site of the C1 gene. These iteron sequences were also found in ToMoTV, exhibiting just few base substitutions. Furthermore, the iteron recognition domain (IRD) of the TLV Rep is identical to that of ToMoTV, which is consistent with the similarities found between the iterated sequences in these species.
Intriguingly, TLV was not found to cause symptoms in field-grown tomato plants, unlike ToMoTV, which has a severe impact on the tomato phenotype. In fact, tomato and benthamiana plants agro-infected with TLV in a controlled environment also remain asymptomatic, while ToMoTV induces a symptomatic phenotype just 10 days after agro-infection of tomato seedlings (these results have been included in a second manuscript).These observations further motivate new studies on their spread and pathogenicity in the island as well as in surrounding regions”.
Discussion lacks key comparisons. The discussion would be significantly strengthened by acknowledging and citing existing literature on LAMP-based detection of TYLCV, a closely related virus. The authors must use this comparison to precisely articulate the specific advance or novel contribution of their present work.
Answer: New text was included:
“For the case of TYLCV, LAMP proved useful for laboratory testing as well as for field diagnosis, even without the need for DNA extraction, using the immunocapture procedure followed by the DNA amplification reaction (24), or using enhanced detection of viral amplicons by a Cas12-directed fluorescent reporter (25). A diagnostic kit is also available from NIPPON GENE CO., LTD (2022), which includes primers targeting a conserved region of the TYLCV genome, enabling the detection of a broad range of TYLCV strains, based on a previous key publication (26). Its application also allows for rapid visualization of the genome's DNA by colorimetric methods (24), or by combining the LAMP assay with real-time methods, its sensitivity could be improved (23). In all these examples, the potential and flexibility of the LAMP procedure have been demonstrated”.
Unsubstantiated claim and internal inconsistency. The claim that "diagnostic tools for ToMoTV and TLV remain unavailable" (Lines 71-72) is not sufficiently supported. The prior mention of TLV detection in Cuban fields and the availability of its complete genome in GenBank contradict this statement, as these facts demonstrate that molecular detection is feasible. The authors must either provide evidence for this claim or revise the text for accuracy.
Answer: New text was included replacing "diagnostic tools for ToMoTV and TLV remain unavailable":
In the case of ToMoTV and TLV, the development of widely used diagnostic and viral tracking technologies has lagged behind that of TYLCV.
Minor comments:
Line 35 ‘Geminiviridae’ should be written in italic. Done
Lines 54 -55 The virus name should not be capitalized. Please double-check this throughout the text. Done
MATERIALS AND METHODS
Primer Design: The manuscript does not explain the rationale for selecting the C3, AV1, BC1, and C1 ORF regions as amplification targets. In particular, the choice of these relatively non-conserved regions requires justification. The authors should elaborate on the specificity of these regions for virus identification and discuss their degree of sequence variability within the Begomovirus.
Answer: The primers were designed to avoid cross amplifications, as justified in the Results section as follows:
“Primers spanning conserved regions of the begomovirus family were excluded to avoid cross-amplification. The selected primers targeted the C3 ORF of TYLCV, the AV1 ORF of the ToMoTV DNA-A component, the BC1 ORF of the ToMoTV DNA-B component, and the C1 ORF of TLV.”
The percentage of identity of each viral target region was not included in the text because it could lead to confusion about the actual feasibility of using it as a target for primer design. When comparing different regions of the genome, irregular but frequent SNPs were observed, which would affect target selection, regardless of the degree of identity between the compared sequences.
Section 2.4 (Field Trials) should be relocated to the end of the M&M section and renumbered as Section 2.7. The preceding subsections should be shifted accordingly. Done
Field sample validation: To more robustly validate the reliability of the established LAMP assay, it is recommended to increase the number of field-collected samples tested.
Answer: We do consider a higher number of samples tested in the field experiment could have been more informative. We are currently requesting authorizations to implement the LAMP method for routine use in evaluating field plants, and look forward to employ it during spring-fall season in 2026.
Template concentration consistency: The template input is inconsistently reported: both PCR and LAMP used 100 ng of DNA, but the total reaction volumes differ (e.g., PCR was performed in 25 µL, whereas the LAMP volume was not specified). This discrepancy leads to a two-fold difference in template concentration, which compromises the fairness of sensitivity comparisons between the two methods. It is recommended that the authors clearly state the template concentration in consistent units (e.g., ng µL⁻¹) and ensure equivalent reaction conditions for an accurate comparison.
Answer: LAMP reaction volume in the manuscript is now specified as 12.5 µl. The volume for PCR was missing, and it was just added in MM section as 12.5 µl during this revision.
RESULTS
Image clarity and presentation: Figures 2–7 are not sufficiently clear and should be replaced with higher-resolution images. In addition, the electrophoresis lanes currently split across Figures 2–4 should be consolidated into a single composite image. The figure legend should clearly annotate what each lane represents.
Answer: Figures 2-7 and legends were revised and improved according to your valuable comments.
Electrophoresis image quality: The resolution of the electrophoresis images is inadequate. In Figures 2A and 2B, the upper fragments of the LAMP ladder exceed 2 kb, indicating possible overloading or improper separation. It is recommended to repeat the electrophoresis using a DNA ladder more appropriate for this fragment size range.
Answer: We are now providing images with higher resolution. We agree LAMP concatamers are large polymers. Currently, we purify Bst polymerase and DNA amplified by LAMP could reach up to 20 Kb. Based in the general opinion, it is important to resolve low molecular weight concatamers to evaluate LAMP reactions.
Specificity testing is incomplete: The specificity evaluation in Figure 3 is limited to cross-testing with single-infection DNA samples and does not include testing with field samples containing dual or triple infections. This oversight leaves open the possibility of false negatives due to template competition. To address this, the authors should include DNA extracted from a sequenced-confirmed triple-infected field sample as a template in the LAMP assay.
Answer: We have focused this assay on demonstrating the absence of cross-reactivity using single-infections as recommended in the literature. For instance, in a parallel assay (cited in this manuscript, Tapia-Sidas et al., 2023), CDC requested to test independent, single viruses to demonstrate specificity. However, we agree a more complex matrix would reinforce the current conclusion of specificity we have routinely achieved.
Limit of detection (LOD) interpretation is overly subjective. The LOD determination in Figure 7 is based solely on visual color change, which is subjective and not corroborated by gel electrophoresis. To improve objectivity, the authors should indicate—either in the figure or the legend—at which template concentrations a band was visible by electrophoresis even in the absence of a color change.
Answer: We appreciate very much your observation on LOD, figure 7 and its legend display the number of template copies; however, color changes in low copy number could be challenging. We consider the LAMP using fluorochromes should increase sensitivity. We look forward to implement such detection as already done in other examples in our facility.
DISCUSSION
Lack of experimental support for pathogenicity in mixed infections: While the authors speculate that "synergism or competition" may underlie the observed symptom differences, this hypothesis lacks experimental validation through inoculation studies using infectious clones. To strengthen their argument, the authors should explicitly frame this as a proposed mechanism and recommend its verification in future research.
Answer: the text was modified as follows:
We all agree this speculation on synergism, competition, or altered plant–vector dynamics is still an open question. The mechanisms underground these mixed infections are our subject of study in the near future, as we consider to have developed the technical resources to detect the viruses under controlled viral inoculation processes using infectious clones.
Absence of comparison with commercially available TYLCV LAMP kits: Although the authors acknowledge the existence of commercial LAMP reagents for TYLCV detection, no direct comparison was made with their newly developed primers. This omission undermines claims regarding the superiority or competitiveness of their method. The authors should either include such a comparative analysis or provide a clear justification for its exclusion (e.g., restrictions on reagent availability in Cuba).
Answer: The main goal of our proposal was to achieve the diagnosis of the three viruses in the field in our specific conditions, using different set of primers but the same basic available reagent tools. The use of commercial kits from overseas are currently restricted. This is the goal of the present paper, to develop a detection method using the available resources. In anear future, the production of in-house enzymes will help us to solve the problem of reagent availability
Underdeveloped discussion of LAMP applicability: The study does not adequately address the potential of LAMP for high-throughput applications or simultaneous detection of multiple viruses. To enhance the impact and relevance of the work, the authors should expand the discussion to include the feasibility of developing multiplex LAMP assays or integrating this method into microfluidic platforms for broader diagnostic use.
Answer: Text was added:
Given the reality of the more frequent occurrence of mixed infections alongside environmental changes and their effects on living organisms, the feasibility of developing and implementing multiplex LAMP is being promoted such as multiplexing LAMP assays, including new technologies based on microfluidic platforms for broader diagnostic use.
REFERENCES
The references are not formatted according to the journal's required style. The authors should carefully revise all citations and the reference list to ensure full compliance with the journal's guidelines.
Already done
Lines 204-210
Figure 4 lacks essential annotations: the molecular weight marker is not labeled with "bp," and there is a discrepancy between the lane numbering in the text (referring to lanes 1–4) and the actual figure (labeled 1, 2, 3, 4). The authors should ensure consistency between the figure and its description.
Answer: revised and corrected
Round 2
Reviewer 2 Report
Comments and Suggestions for Authors
The authors have largely addressed the previous review comments. However, I have three additional suggestions to further improve the manuscript:
If feasible, I recommend increasing the sample size used in the practical application section to more robustly demonstrate the method's effectiveness.
Given the high sensitivity of LAMP assays and their susceptibility to false positives (e.g., from aerosol contamination), please include a discussion on these risks and the specific mitigation strategies employed.
The figure quality remains insufficient. Please replace the current figures with high-resolution images. Specifically, for Figure 1, please provide the original, uncropped electrophoresis images. The lanes appear to be spliced or artificially grouped; to ensure data integrity and confirm that samples were run on the same gel, the full gel should be shown with all lanes clearly annotated.
Author Response
The authors have largely addressed the previous review comments. However, I have three additional suggestions to further improve the manuscript:
If feasible, I recommend increasing the sample size used in the practical application section to more robustly demonstrate the method's effectiveness.
Dear reviewer,
We thank you very much for your valuable observations. We look forward to test tomato plants from the field to get a practical demonstration. In this very moment, we do not have access to naturally infected plants, however, we are ready to assay them, as we have already purified Bst enzyme and primers have been tested with positive controls. Our goal is to measure virus dynamics and their possible interactions in 2026.
Given the high sensitivity of LAMP assays and their susceptibility to false positives (e.g., from aerosol contamination), please include a discussion on these risks and the specific mitigation strategies employed.
The following paragraph was inserted:
Although the LAMP diagnostic tool is sensitive and enables early detection of target nucleic acids, the mandatory use of both positive and negative controls is strongly recommended. Field use of portable devices may lead to false-positive results, likely due to environmental contamination. Therefore, any positive result obtained from a field sample should be confirmed in the laboratory using good laboratory practices (e.g., decontaminated work surfaces and dedicated, decontaminated pipettes).
The figure quality remains insufficient. Please replace the current figures with high-resolution images. Specifically, for Figure 1, please provide the original, uncropped electrophoresis images. The lanes appear to be spliced or artificially grouped; to ensure data integrity and confirm that samples were run on the same gel, the full gel should be shown with all lanes clearly annotated.
Please find the raw, uncropped electrophoresis images, showing every step to conform the Figure 1. You will find attached the pdf file: Figure 1, step by step.
We all appreciate your careful review, thank you kindly.
